# Expression Dynamics of *lpa1* Gene and Accumulation Pattern of Phytate in Maize Genotypes Possessing *opaque2* and *crtRB1* Genes at Different Stages of Kernel Development

**DOI:** 10.3390/plants12091745

**Published:** 2023-04-24

**Authors:** Vinay Bhatt, Vignesh Muthusamy, Kusuma Kumari Panda, Ashvinkumar Katral, Rashmi Chhabra, Subhra J. Mishra, Ikkurti Gopinath, Rajkumar U. Zunjare, Chirravuri Naga Neeraja, Sujay Rakshit, Devendra K. Yadava, Firoz Hossain

**Affiliations:** 1Division of Genetics, ICAR—Indian Agricultural Research Institute, New Delhi 110012, India; 2AMITY Institute of Biotechnology, AMITY University, Noida 201313, Uttar Pradesh, India; 3ICAR—Indian Institute of Rice Research, Hyderabad 500030, India; 4ICAR—Indian Institute of Maize Research, Ludhiana 141004, India; 5ICAR—Indian Institute of Agricultural Biotechnology, Ranchi 834010, India

**Keywords:** low phytic acid, expression, gene transcript, minerals, *Zea mays*

## Abstract

Phytic acid (PA) acts as a storehouse for the majority of the mineral phosphorous (P) in maize; ~80% of the total P stored as phytate P is not available to monogastric animals and thereby causes eutrophication. In addition, phytic acid chelates positively charged minerals making them unavailable in the diet. The mutant *lpa1-1* allele reduces PA more than the wild-type *LPA1* allele. Further, mutant gene *opaque2* (*o2*) enhances lysine and tryptophan and *crtRB1* enhances provitamin-A (proA) more than wild-type *O2* and *CRTRB1* alleles, respectively. So far, the expression pattern of the mutant *lpa1-1* allele has not been analysed in maize genotypes rich in lysine, tryptophan and proA. Here, we analysed the expression pattern of wild and mutant alleles of *LPA1*, *O2* and *CRTRB1* genes in inbreds with (i) mutant *lpa1*-*1*, *o2* and *crtRB1* alleles, (ii) wild-type *LPA1* allele and mutant *o2* and *crtRB1* alleles and (iii) wild-type *LPA1*, *O2* and *CRTRB1* alleles at 15, 30 and 45 days after pollination (DAP). The average reduction of PA/total phosphorous (TP) in *lpa1-1* mutant inbreds was 29.30% over wild-type *LPA1* allele. The *o2* and *crtRB1*-based inbreds possessed ~two-fold higher amounts of lysine and tryptophan, and four-fold higher amounts of proA compared to wild-type alleles. The transcript levels of *lpa1-1*, *o2* and *crtRB1* genes in *lpa1-1*-based inbreds were significantly lower than their wild-type versions across kernel development. The *lpa1-1*, *o2* and *crtRB1* genes reached their highest peak at 15 DAP. The correlation of transcript levels of *lpa1-1* was positive for PA/TP (r = 0.980), whereas it was negative with inorganic phosphorous (iP) (r = −0.950). The *o2* and *crtRB1* transcripts showed negative correlations with lysine (r = −0.887) and tryptophan (r = −0.893), and proA (r = −0.940), respectively. This is the first comprehensive study on *lpa1-1* expression in the maize inbreds during different kernel development stages. The information generated here offers great potential for comprehending the dynamics of phytic acid regulation in maize.

## 1. Introduction

Mineral deficiencies cause severe health concerns including stunted development, prenatal problems, intellectual disabilities and an increased risk of morbidity and death in humans [1]. These minerals are essential for healthy cell development and signalling [2,3]. Population growth and natural disasters warrant demand for nutrient-rich foods which escalated after the COVID-19 pandemic [4]. Addressing the problem of malnutrition through biofortified staple food is a more affordable and sustainable option than food fortification and medical supplementation [5,6].

Phytic acid (PA), also known as myo-inositol 1,2,3,4,5,6-hexakisphosphate, is the main phosphorous (P) storage molecule in seeds [7]. The negatively charged PA chelates positively charged minerals, making them inaccessible to the human gut [8,9]. Phosphorous is accumulated in growing seeds in excess of what is required for normal cellular function during germination. Monogastric animals are unable to digest PA, which is passed through excretion leading to environmental pollution [10,11]. The rise in phosphorous levels in water bodies due to release of PA from animals such as poultry, results in eutrophication or excessive algae growth [12,13]. Therefore, reduction of PA not only improves the nutritional quality of food, but also alleviates malnutrition in addition to reducing the environmental worries [10,14].

Maize (*Zea mays* ssp. *mays*) possesses considerable significance as feed, food and raw material in a variety of industrial purposes and is at the centre of a global initiative to address micronutrient deficiency in underdeveloped nations through biofortification [15]. In order to increase intake of high-quality proteins, quality protein maize (QPM) cultivars are nutritionally enhanced. High lysine, tryptophan and provitamin-A and less anti-nutritional factors, make these maize genotypes nutritionally superior. As compared to other cereals, maize grains have a higher content of PA, which significantly lowers the bioavailability of minerals [14,16]. Although wide genetic variation for mineral content in maize germplasms have been recorded, polygenic nature and strong environmental influence have limited the development of mineral-rich cultivars [16,17]. Several low phytic acid (*lpa*) mutants viz. *lpa1, lpa2, lpa3, lpa1-7* and *lpa241* have been reported in maize [18,19,20,21]. Of these, *lpa1-1* mutation with a modification in the trans-membrane transporter protein reduces PA by 55–65% and is not associated with harmful effects on seed germination or seedling vigour [19,22,23,24,25]. Ragi et al. [25] evaluated the *lpa1-1*-based maize genotypes at multiple locations and observed ~35% reduction in phytic acid in the mutant genotypes. They also reported similar agronomic performance of the *lpa1-1*-based mutant genotypes as compared to the wild-type genotypes.

Breeding efforts at ICAR-Indian Agricultural Research Institute (IARI), New Delhi, have produced a number of sub-tropically adapted *lpa1-1*-based maize inbreds through marker-assisted selection (MAS) [24,25]. Further, traditional maize contains unbalanced protein which is poor in essential amino acids (lysine and tryptophan) and provitamin-A (proA) [26]. Mutant *opaque2* (*o2*) and *crtRB1* genes enhance the lysine, tryptophan and proA in maize [27]. Though significant progress has been made to develop sub-tropically adapted nutritionally rich maize genotypes, no information is available on the expression pattern of *lpa1-1* gene at different stages of kernel development. Further, studying the expression of *o2* and *crtRB1* in *lpa1-1*-based genotypes would provide an understanding of the combined effects of these genes on different nutritional quality parameters at different kernel development stages. Furthermore, the transcript expression study would shed light on the intricate regulation of these genes in the biosynthesis pathways, regulate micronutrient bioavailability, and help breeders to determine the precise stage of maize kernel harvest. Early stages of maize are the green forms that we can utilise as a green cob (green ears) which is very popular in Asia, whereas mature stages of maize can be utilised in the form of flour, chapattis and porridges. Hence, we targeted 15, 30 and 45 days after pollination (DAP) for the experiment. With this experiment, we can detect the precise stage of harvest where people can utilise the most nutrients available. The present study was therefore undertaken to analyse the (i) accumulation of PA, inorganic phosphorous (iP) and PA/total phosphorous (TP), (ii) expression pattern of the *lpa1-1, o2* and *crtRB1* genes, and (iii) correlation among the transcript levels and nutrient accumulation at different stages of kernel development.

## 2. Results

### 2.1. Genetic Variation for Nutritional Quality Traits

The results of ANOVA revealed significant differences (*p* < 0.01) among the genotypes and kernel developmental stages for PA, iP, TP, PA/TP, lysine, tryptophan and proA (Table 1). The interaction of genotype (G) × days after pollination (DAP) also showed significant (*p* < 0.01) differences for all the nutritional traits (Table 1). The contribution of genotypes for the total variation was the highest (range: 54.67 to 80.16%) followed by DAP (range: 11.72 to 64.58%) and G × DAP (range: 0.30 to 12.10%) across the nutritional quality traits. The PA, iP and PA/TP varied from 1.08 to 2.84 mg/g, 0.22 to 1.12 mg/g and 50.30 to 92.70%, respectively (Appendix A). Across genotypes, lysine, tryptophan and proA ranged from 0.149 to 0.564%, 0.035 to 0.161% and 1.61 to17.26 µg/g, respectively (Appendix A).

### 2.2. PA and Associated Traits during Different Stages of Kernel Development

The PA among the *lpa-1-1*-based inbreds ranged from 1.08 to 1.87 mg/g across 15, 30 and 45 DAP, whereas it varied from 2.10 to 2.84 mg/g among the wild-type inbreds (Appendix A; Figure 1a). An average reduction of PA among *lpa1-1*-based inbreds was 41.68% as compared to wild-type inbreds. Likewise, the iP among *lpa1-1*-based inbreds varied from 0.91 to 1.12 mg/g with the mean of 1.0 mg/g (Appendix A; Figure 1b). However, the mean iP among the wild-type inbreds was 0.50 mg/g with a range of 0.22 to 0.90 mg/g across 15, 30, and 45 DAP. Among the *lpa1-1*-based inbreds, nearly two-fold increment in iP content was observed over the wild-type inbreds. Mean PA content of *lpa1-1*-based inbreds ranged from 1.13 mg/g (15 DAP) and 1.77 mg/g (45 DAP). Similar patterns of PA were observed for wild-type inbreds as well. Among the *lpa1-1*-based inbreds, highest mean iP was observed at 15 DAP (1.03 mg/g) and lowest iP at 45 DAP (0.98 mg/g). In contrast, significant difference of iP was observed among wild-type inbreds with highest iP at 15 DAP and the lowest iP at 45-DAP. The range of PA/TP ratio varied from 50.30 to 66.00% with a mean of 58.73% among *lpa1-1*-based inbreds (Appendix A; Figure 1c). However, the PA/TP ratio among the wild-type (*LPA1-1/LPA1-1*) inbreds varied from 70.60 to 92.70% with a mean of 83.00%. The mean PA/TP ratio among *lpa1-1*-based inbreds showed the lowest at 15 DAP (52.60%) and highest at 45-DAP (64.40%). Across the wild-type inbreds, the increasing trend was observed in PA/TP ratio with lowest ratio at 15 DAP (74%) and highest ratio at 45 DAP (91%). The average reduction of PA/TP in *lpa1-1* introgressed inbreds was 29.30% as compared to their wild-type versions. The *lpa1-1*-based inbreds (PMI-PV5-*lpa1*, PMI-PV6-*lpa1*, PMI-PV7-*lpa1* and PMI-PV8-*lpa1*) revealed reduction in mean PA/TP ratio of 23.40%, 26.66%, 23.68% and 24.40% over their respective original versions, viz: PMI-PV5, PMI-PV6, PMI-PV7 and PMI-PV8 (Figure 1c).

### 2.3. Lysine and Tryptophan during Different Stages of Kernel Development

The *lpa1-1* progenies with *o2* allele possessed lysine content ranging from 0.329 to 0.564% (Figure 2a), while tryptophan content ranged from 0.081 to 0.161% (Figure 2b) across different stages of kernel development (Appendix A). The mean lysine (0.424%) and tryptophan (0.111%) were higher than that of the control genotype, HKI1105 with wild-type *O2* allele (lysine: 0.187%, tryptophan: 0.051%) across all stages of kernel development. The *lpa1-1*-based progenies showed the highest mean lysine (0.510%) and tryptophan (0.147%) at 15 DAP followed by 0.402% and 0.098% at 30 DAP, which further decreased to 0.358% and 0.087% at 45 DAP, respectively. The inbreds, viz. PMI-PV5, PMI-V6, PMI-PV7 and PMI-PV8, showed a similar pattern with high lysine content (0.424%) and tryptophan content (0.108%) at 15 DAP due to the presence of the mutant *o2* alleles. Similarly, HKI1105 possessing wild-type *O2* allele also possessed the highest mean values of lysine (0.232%) and tryptophan (0.066%) at 15 DAP and lowest mean values of lysine (0.149%) and tryptophan (0.035%) at 45 DAP (Appendix A).

### 2.4. Provitamin-A during Different Stages of Kernel Development

Provitamin-A among the *lpa1-1*-based inbreds with mutant allele of *crtRB1* ranged from 9.28 to 17.26 µg/g (Appendix A; Figure 2c). In comparison, HKI1105 with wild-type *CRTRB1* allele possessed very low proA (3.05 µg/g) over the *lpa1-1*-based mutant lines (12.99 µg/g) across the kernel developmental stages. The *lpa1-1*-based inbreds showed highest mean proA at 15 DAP (15.99 µg/g) followed by 30 DAP (12.24 µg/g) and 45 DAP (10.73 µg/g). Since PMI-PV5, PMI-PV6, PMI-PV7 and PMI-PV8 possessed the *crtRB1* mutant allele, their progenies exhibited a similar pattern and higher mean value as *lpa1-1* introgressed individuals. A similar trend was observed in the case of control (HKI1105) as well; it possessed the wild-type allele with highest proA of 5.08 µg/g at 15 DAP followed by 2.48 µg/g at 30 DAP and lowest mean of 1.61 µg/g at 45 DAP.

### 2.5. Variation in Expression of Genes

ANOVA revealed that transcript levels of *lpa-1-1, o2* and *crtRB1* genes revealed significant differences (*p* < 0.01) among the genotypes as well as among the different stages of kernel development (Table 2). The expression level of *lpa1-1* transcripts (2^−ΔC^_T_) varied from 0.064 to 2.121 across the genotypes, while it was 0.011 to 0.821 and 0.010 to 0.387 for *o2* and *crtRB1* genes, respectively (Table 3).

### 2.6. Expression Pattern of lpa-1-1 Gene during Kernel Development Stages

The highest expression level of *lpa-1-1* gene was observed at 15 DAP and lowest expression was observed at 45 DAP (Table 3; Figure 3a). The *lpa1-1*-based inbred progenies (PMI-PV5-*lpa1*, PMI-PV6-*lpa1*, PMI-PV7-*lpa1* and PMI-PV8-*lpa1*) revealed the transcript levels of 4.20, 3.63, 4.36 and 4.10-fold lower than their original versions, viz.: PMI-PV5, PMI-PV6, PMI-PV7 and PMI-PV8, respectively. Across the introgressed lines, *lpa1-1* mutant allele possessed four-fold lower transcript levels over the wild-type inbreds having *LPA1* allele (Table 3).

### 2.7. Expression Pattern of o2 Gene during Kernel Development Stages

The *lpa1-1* inbreds with mutant *o2* allele and HKI1105 possessing wild-type *O2* allele showed differences in transcript levels of *o2* gene during different stages of kernel development (Figure 3b). The highest mean expression of *o2* gene was observed at 15 DAP (0.067) followed by 30 DAP (0.028), while the lowest expression level was detected at 45 DAP (0.017) across the *lpa1-1*-based inbreds (Figure 3b). The control inbred HKI1105 showed a similar pattern for *o2,* with highest expression at 15 DAP and lowest expression at 45 DAP. Among the *lpa1-1*-based improved lines, the highest expression level of *o2* was observed for PMI-PV6-*lpa1*-A (0.079) at 15 DAP, while the lowest expression level was found for PMI-PV5-*lpa1*-B (0.012) at 45 DAP. The highest and lowest transcript levels of *O2* in HKI1105 were 0.821 and 0.205, respectively. Across the kernel development stages, *lpa1-1*-based improved lines recorded significantly lower levels of *o2* expression (13.07-fold lower transcripts) over the wild-type (*O2*) in control inbred (HKI1105). The inbreds, viz. PMI-PV5, PMI-PV6, PMI-PV7 and PMI-PV8, carried the mutant *o2* allele, hence the expression pattern observed was similar to *lpa1-1* introgressed progenies having mutant *o2* allele.

### 2.8. Expression Pattern of crtRB1 Gene during Different Stages of Kernel Development

The inbred (HKI1105) with wild-type *CRTRB1* allele and *lpa1-1*-based lines with mutant *crtRB1* allele displayed significant differences in the transcript levels during kernel developmental stages (Figure 3c). The mean expression level of *crtRB1* among *lpa1-1*-based lines attained highest peak at 15 DAP (0.032), while the lowest expression level was noticed at 45 DAP (0.014). The wild-type *CRTRB1* allele in HKI1105 showed a similar trend with highest mean expression level at 15 DAP (0.387) and lowest expression level at 45 DAP (0.209) (Figure 3c). However, the mean expression level was significantly higher in HKI1105 compared to *lpa1-1*-based inbreds (Figure 3c). The progeny, viz. PMI-PV5-*lpa1*-B among *lpa1-1*-based inbreds, showed highest expression of *crtRB1* gene at 15 DAP, while PMI-PV6-*lpa1*-B showed lowest expression at 45 DAP. The *lpa1-1*-based improved lines recorded significantly lower transcript levels of *crtRB1* (12.32-fold) over wild-type *CRTRB1* allele present in HKI1105. The *lpa1-1*-based introgressed lines possessed *crtRB1* mutant allele which was also present in PMI-PV5, PMI-PV6, PMI-PV7 and PMI-PV8; hence, the expression levels were comparable among recurrent parents and *lpa1-1*-introgressed lines, displaying the same expression pattern and mean expression levels (Table 3).

### 2.9. Correlation between Transcript Levels and Accumulation of Nutrients

Strong positive correlation between transcript level of *lpa1-1* with PA (r = 0.98 to 0.99) and PA/TP (r = 0.98 in all three kernel development stages) was observed during different stages of kernel development across all wild-type and mutant versions (Table 4; Appendix A). However, the *lpa1-1* expression showed significantly negative correlation with iP accumulation during their respective kernel developmental stages across all wild-type and mutant versions (r = −0.98 to −0.90) (Appendix A). Similarly, negative correlation was found for transcript levels of *o2* gene with lysine (r = −0.91 to −0.84) and tryptophan (r = −0.92 to −0.85) at various kernel developmental stages (Appendix A). Moreover, the expression level of *crtRB1* gene possessed negative correlation with proA (r = −0.94) in all three stages of kernel development across the genotypes (Appendix A).

## 3. Discussion

Phytic acid (PA) is a critical anti-nutritional factor in human and monogastric animals as negatively charged phosphorous in PA chelates positively charged minerals and makes them unavailable in the gut [11,28]. In addition, traditional maize is poor in proA and essential amino acids such as lysine and tryptophan [27]. Here, we investigated accumulation pattern of PA, proA, lysine and tryptophan, and expression of *lpa1-1, crtRB1* and *o2* genes among a set of newly developed unique maize genotypes during different stages of kernel development [29]. The expression profiling of target genes and the study of their relationships with nutrient accumulation provides new insights into their regulation in the biosynthesis pathway during various kernel developmental stages [30].

### 3.1. Expression Pattern of lpa1-1 and Its Effect on Kernel Phytate Accumulation

Comparing controls with wild-types and *lpa1-1*-based inbreds, we found that the transcript levels in the wild-types were 4.1-fold higher. These comparisons also showed a strong positive relationship of *lpa1-1* transcript levels with accumulation of PA and PA/TP ratio. Low PA and PA/TP ratio signified the higher bioavailability of minerals as compared to wild-type inbreds [24,25]. The *lpa1-1* mutant was developed through ethyl methanesulfonate (EMS) at United States Department of Agriculture (USDA) [18]. Mutation is generated due to distortion in a trans-membrane transporter protein (*ZmMRP4*) gene that encodes for membrane transport protein and helps in transport of PA to storage vacuole [31]. The defective transporter protein is unable to store PA inside protein-storage vacuoles [31]. The maximum expression of *lpa1-1* was observed at 15 DAP, while the least expression was observed at 45 DAP. However, the maximum concentration of PA was observed at 45 DAP and minimum at 15 DAP among *lpa1-1*-based improved lines. In the case of iP, the majority of *lpa1-1*-based lines showed maximum concentration accumulated at 15 DAP and minimum accumulation at 45 DAP. PA/TP ratio showed lowest percentage at 15 DAP and highest at 45 DAP.

### 3.2. Expression Pattern of o2 and Its Effect on Lysine and Tryptophan

The *lpa1-1*-based maize inbreds possessed significantly higher lysine (2.27-fold) and tryptophan (2.17-fold) compared to the wild-type (*O2*) control. In this study, the *lpa1-1*-based inbreds were found to have 13-fold reduced expression of the *o2* allele in comparison to the wild-type *O2* allele in control inbred. Additionally, negative correlation was observed between *o2* transcript levels and accumulation of amino acids (lysine and tryptophan). The leucine zipper family transcriptional factors, which are necessary for the synthesis of 22-kDa-α-zeins (which lack lysine), is encoded by the *O2* gene [32]. The *o2* mutant gene is responsible for enhancing the synthesis of lysine-rich non-zein protein rich in lysine and tryptophan in addition to negatively regulating lysine keto-reductase [33]. The highest mean expression of *o2* gene was observed at 15 DAP, while the lowest mean expression was observed at 45 DAP. The decreasing pattern of amino acids (lysine and tryptophan) with kernel maturity indicates that delay would lead to the loss of nutritional quality [32,34]. As moisture content in maize kernels decreases, genes involved in the production of micronutrients are prevented from being transcribed [35]. This indicated that higher nutritional quality could be achieved when cobs are harvested as green cobs rather than at the dried stage.

### 3.3. Expression Pattern of crtRB1 and Its Effect on proA Accumulation

In the current study, *lpa1-1*-based inbreds with mutant *crtRB1* allele had significantly higher proA (4.3-fold) over those having wild-type *CRTRB1* allele. The *lpa1-1* inbreds had ~12.32-fold lesser transcript levels than the wild-type inbred carrying *CRTRB1* allele at various stages of kernel development. The favourable *crtRB1* allele enhances the proA content by partially blocking the hydroxylation stages in the branch of the carotenoid biosynthesis pathway [36]. This was additionally supported through significant negative correlation between the levels of *crtRB1* transcripts and accumulation of proA carotenoids [37]. The highest expression of *crtRB1* was observed at 15 DAP, while the lowest expression level was at 45 DAP. Maize kernels gradually lose moisture content as they mature, which lowers the transcription of several genes associated with vital nutrients. This suggested that green cobs harvested at dough stage would provide higher proA than those harvested as dried stage.

The study clearly suggests that the maize genotypes developed and evaluated in the present study would offer highly bioavailable phosphorous which would help in growth and development of bones and also prevents eutrophication. In addition, the inbreds rich in lysine, tryptophan and provitamin-A would help in developing maize hybrids for their future use in providing higher protein quality and vitamin-A.

## 4. Materials and Methods

### 4.1. Plant Materials

The plant materials consisted of four *lpa1-1* (mutant allele)-based maize inbreds (*PMI-PV5-lpa1-1, PMI-PV6-lpa1-1, PMI-PV7-lpa1-1* and PMI-PV8-lpa1-1) and their respective wild-type inbreds (*PMI-PV5, PMI-PV6, PMI-PV7* and PMI-PV8). The *lpa1-1*-based inbreds were developed through marker-assisted introgression of *lpa1-1* allele in the background of proA, lysine and tryptophan rich maize inbreds, viz. PMI-PV5, PMI-PV6, PMI-PV7 and PMI-PV8, which possessed *o2* (mutant), *crtRB1* (mutant) and *LPA1* (wild-type) alleles. These inbreds are parents of three released biofortified maize hybrids, viz.: Pusa HQPM1 Improved (PMI-PV7 × PMI-PV6), Pusa HQPM5 Improved (PMI-PV6 × PMI-PV5) and Pusa HQPM7 Improved (PMI-PV7 × PMI-PV5), and a promising hybrid, Pusa HQPM4 Improved (PMI-PV8 × PMI-PV5). Two progenies (-A and -B) in each genetic background of *lpa1-1*-based inbreds and their wild-type inbreds were analysed for the differential expression of *lpa1-1*, *o2* and *crtRB1* genes. The inbred HKI1105, used as control, possessed wild-type alleles (*O2*, *CRTRB1* and *LPA1*) of all three genes.

### 4.2. Field Experiments

A set of 13 selected inbreds with (i) mutant *lpa1*-*1*, *o2* and *crtRB1* alleles, (ii) wild-type *LPA1* allele and mutant *o2* and *crtRB1* alleles, and (iii) wild-type *LPA1*, *O2* and *CRTRB1* alleles, were evaluated at ICAR-IARI, New Delhi (29°41′52.13″ N, 77°0′24.95″ E) during the rainy season (July–October) of 2022. Two replications of each entry were raised in one row of 3.0 m length following randomised complete block design (RCBD) with row-to-row and plant-to-plant spacing of 75 cm and 20 cm, respectively. Standard agronomic practices were followed to raise the crop as described in earlier studies [33]. Three to four plants in each of the genotypes per replication were selfed to avoid xenia effects caused due to foreign pollens. The selfed-ears were harvested at 15, 30 and 45 days after pollination (DAP) from each replication. The selfed-ears were stored at −80 °C until RNA isolation and determination of quality parameters.

### 4.3. Isolation of RNA and cDNA Synthesis

RNA was isolated from kernels following a modified version of the traditional RNA Trizol technique as per Dutta et al. [38]. RNA quality was checked using 0.8% agarose gel with 1% formaldehyde and Nabi UV/Vis Nano Spectrophotometer by determining the ratio of absorbance at 260/280 nm. The total RNA was converted to cDNA through reverse transcription using Verso cDNA Synthesis Kit (Thermo Fisher Scientific Baltics UAB, Lithuania, Vilnius). For each reaction, 1 µL of Verso Enzyme Mix, 2 µL of dNTP mix, 1 µL of anchored oligo-dT, 1 µL of RT enhancer, 4 µL of 5X cDNA synthesis buffer, 5 µg of template (RNA) and nuclease-free water were combined to make a 20 µL reaction mixture. BIO-RAD T100^TM^ thermal cycler (Bio-Rad Laboratories, Inc., Singapore) was used for incubation to convert to cDNA at 42 °C for 60 min followed by 95 °C for 2 min. A UV-spectrophotometer was used to measure the concentration of cDNA samples.

### 4.4. Designing of Primers for the Expression Studies

Primer3 Input v4.1.0 software was used to design the primers for expression of *lpa1-1*, *o2*, *crtRB1* and *Adh1* (endogenous control) with GenBank accession numbers EF586878, X15544, GQ889716 and NC 050096.1, respectively. *lpa1-1* is a mutant of *Lpa1* gene that encodes for a multidrug resistance-associated protein (MRP) ATP-binding cassette (ABC) transporter responsible for transporting phytic acid to the storage vacuole [31]. In maize kernels, *CrtRB1* encodes for *β-carotene hydroxylase*, which converts α- and β-carotene into non-provitamin A carotenoids, viz. lutein and zeaxanthin [36]. Leucine-zipper (bZIP) protein, which is produced by the *O2* gene, functions as a transcriptional factor for development of the zein family of storage protein genes, particularly 22 kDa α-zein in maize [39]. *Alcohol dehydrogenase 1 (ADH1)* is a constitutive gene that aids in cell survival in low oxygen environments (anaerobic conditions) [40]. Details of the expression primers used are given in Appendix A. The primers were synthesized by M/S Macrogen Pvt. Ltd., Seoul, Korea.

### 4.5. Gene Expression Analysis for lpa1-1, o2 and crtRB1 Genes

For each cDNA sample across the three dates of harvest, the chosen primer pairs for *lpa1-1, o2* and *crtRB1* as well as the *Adh1* endogenous control gene were standardised for real-time polymerase chain reaction (RT-PCR) analysis in duplicate by mixing varied forward and reverse primer concentrations ranging from 62.5 to 1000 nM using CFX OPUS 96 Real-time PCR System (BIO-RAD) instrument. The optimised concentration of each primer pair for *lpa1-1, o2, crtRB1* and *Adh1* were (i) forward primer: 0.25 μM, reverse primer: 1μM, (ii) forward primer: 0.25 μM, reverse primer: 0.5 μM, (iii) forward primer: 1 μM, reverse primer: 0.25 μM, and (iv) forward primer: 0.5 μM, reverse primer: 1 μM, respectively. The 20 μL reaction mixture for RT-PCR contained the following components: 100 ng of template cDNA, 1X EvaGreen, 1X ROX and optimised concentration of forward and reverse primers. The PCR conditions followed an initial denaturation of hold stage at 95 °C for 5 min; PCR stage of 55 cycles of 95 °C for 10 s, 60 °C for 15 s and 72 °C for 20 s; and melt curve from 65 °C to 95 °C with 0.5 °C temperature increment after every 5 s on continuous mode. Each sample’s C_T_ value was obtained through RT-PCR and the transcript level of each gene was determined using the 2^−ΔC^_T_ method. The expression level of *lpa1-1* in inbreds (*PMI-PV5-lpa1-1, PMI-PV6-lpa1-1, PMI-PV7-lpa1-1* and PMI-PV8-lpa1-1) possessing the *lpa1-1*, *o2* and *crtRB1* alleles was compared with their original versions, viz.: PMI-PV5, PMI-PV6, PMI-PV7 and PMI-PV8 (possessing the *LPA1-1*, *o2* and *crtRB1* alleles). Whereas, the *lpa1-1*, *o2* and *crtRB1* gene expression levels were compared to that of HKI1105 (possessing wild-type *LPA1-1*, *O2* and *CRTRB1* alleles).

### 4.6. Samples Preparation for Biochemical Analysis

Part of the self-pollinated ears (harvested at 15, 30 and 45 DAP) remaining after RNA isolation were shelled and left to dry for a further period of 72 h at room temperature in a silica gel-filled container [41]. Randomly selected kernels (n = 100) were taken and kept in desiccator with silica gel particles replaced in every 24 h to keep the kernel moisture content below 15%. The container was completely covered, and the kernels were wrapped in blotting paper using aluminium foil to avoid deterioration brought on by light compounds. The drying process was carried out at a temperature of 25 °C and the dried kernels were processed into a fine powder in the dark using a coffee grinder (Philips, HL7756) machine. Samples from each of the genotypes were reduced to 5 g using quartering technique and stored at −20 °C for further analysis.

### 4.7. Determination of Phytic Acid (PA) and Inorganic Phosphorous (iP)

A modest modification of the protocol published by Lorenz et al. (2007) was followed for the determination of PA and iP [42]. In 2 mL micro-centrifuge tube, 100 mg of maize kernel powder from each of the samples was added along with 2 mL of 0.65 M HCl. The tubes were then placed on a shaker with 200 rpm for an overnight period at room temperature and then centrifuged for 5 min at 10,000 rpm. From the extract of each of the samples, 500 µL was transferred to a new 15 mL tube for iP quantification and another 500 µL to a 2 mL micro-centrifuge tube for quantification of PA. Quantitative standards for PA and iP were used in equal amounts. KH_2_PO_4_ (HiMedia, Thane West, Maharashtra, India) and phytic acid dodecasodium salt from maize (Sigma-Aldrich Chemicals Private Ltd., Industrial Area, Bangalore, India) were used as iP standard and PA standard, respectively. Reagent for the iP quantification was prepared right before use and contained in the ratio of 2 (double distilled H_2_O): 1 (ammonium molybdate (0.02 M)): 1 (sulphuric acid (3 M)): 1 (ascorbic acid (0.57 M)). For the determination of iP, each sample received one ml of reagent and one ml of double-distilled water. The optical density was measured at 820 nm after 15 min of incubation at room temperature when the blue colour had fully developed. In order to determine the amount of PA, 1.25 mL of the wade reagent was added to each micro-centrifuge tube. The pink colouration developed after an incubation period of 15 min under room temperature and optical density was measured at 490 nm. The wade reagent consisted of 80 mL of double-distilled water, 0.03 g of FeCl_3_·6H_2_O and 0.3 g of 5-sulfosalicylic acid. The aforementioned solution was chilled overnight, and the following day, it was treated with NaOH to bring its pH level to 3.05 and volume made up to 100 mL. PA was divided by conversion factor 3.55 to convert to phytate phosphorous (P) [24]. Proportion of PA in the kernel was determined as the ratio of phytate (PA)/TP, where TP = PA + iP, and the ratio was converted to percent.

### 4.8. Determination of Lysine, Tryptophan and Provitamin-A

Dried kernels from stored kernels were analysed for lysine, tryptophan and proA using UHPLC system (Ultra High-Performance Liquid Chromatography; Thermo Scientific, Massachusetts, USA) [43,44] Thermo Scientific’s Acclaim-120 C_18_ (5 µm, 120Å, 4.6 × 150 mm) column was used to separate samples of lysine and tryptophan, which were then analysed using a diode array detector-3000 (RS) with absorbance at 265 nm and 280 nm, respectively. By comparing the area of the amino acid mix standard with the area of the sample, the concentration of lysine and tryptophan present in the sample were determined. To avoid degradation due to oxidation by light, carotenoids extraction was carried out in the dark, using a modified version of a previously reported procedure [45]. Specimens were taken using YMC Carotenoid C_30_ column (5 m, 4.6 × 250 mm) through UHPLC system (Thermo Scientific, Walthamm, MA, USA). A diode array detector-3000 (RS) with absorbance at 450 nm was used to detect β-carotene (BC) and β-cryptoxanthin (BCX). ProA concentration was calculated as the sum of BC and 50% of BCX [27].

### 4.9. Statistical Analysis

Mean nutritional quality traits and transcript levels at different kernel developmental stages were considered for analysis of variance (ANOVA) using ‘anova’ function in ‘R v4.2.1′ statistical tool. The correlation plots were deduced using ‘metan’ package in ‘R v4.2.1′. The bar plots were constructed using Microsoft Office-Excel 2019.

## 5. Conclusions

Phytic acid in maize acts an anti-nutritional factor reducing the bioavailability of mineral nutrients. The results revealed that the introgression of favourable allele of *lpa1-1* led to significant reduction of PA and PA/TP ratio and enhancement in iP. Favourable *o2* allele led to enhancement in lysine and tryptophan, while *crtRB1* introgression resulted in enhancement in proA. The transcript levels of *lpa1-1, o2* and *crtRB1* genes were significantly lower than wild-type alleles (*LPA1, O2* and *CRTRB1*). The highest reduction of PA was observed at 15 DAP which also had the highest accumulation of proA, lysine and tryptophan. The transcript levels of *lpa1-1* gene showed positive correlation with PA, while negative correlation was found for *o2* with lysine and tryptophan, and *crtRB1* with proA. This is the first comprehensive study on understanding the regulation pattern of *lpa1-1*, *o2* and *crtRB1* genes and their relationship with accumulation of PA, iP, PA/TP, lysine, tryptophan and proA at different stages of kernel development among *lpa1-1*-based maize genotypes.

## Figures and Tables

**Figure 1 plants-12-01745-f001:**
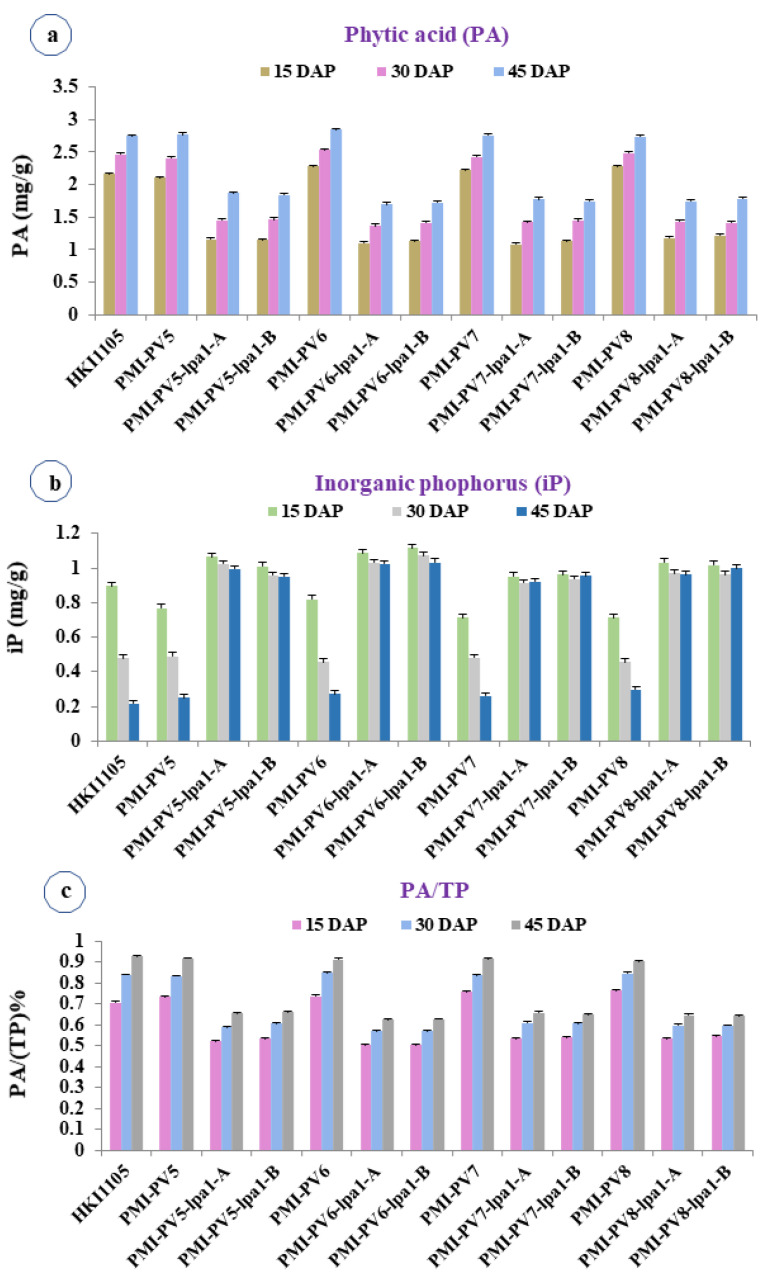
Mean concentration of nutritional parameters at different kernel developmental stages. (**a**) PA, (**b**) iP, (**c**) PA/TP. PA/TP: Phytic acid/Total phosphorous (%); DAP: Days after pollination. Error bars represents the standard error at 5% level of significance.

**Figure 2 plants-12-01745-f002:**
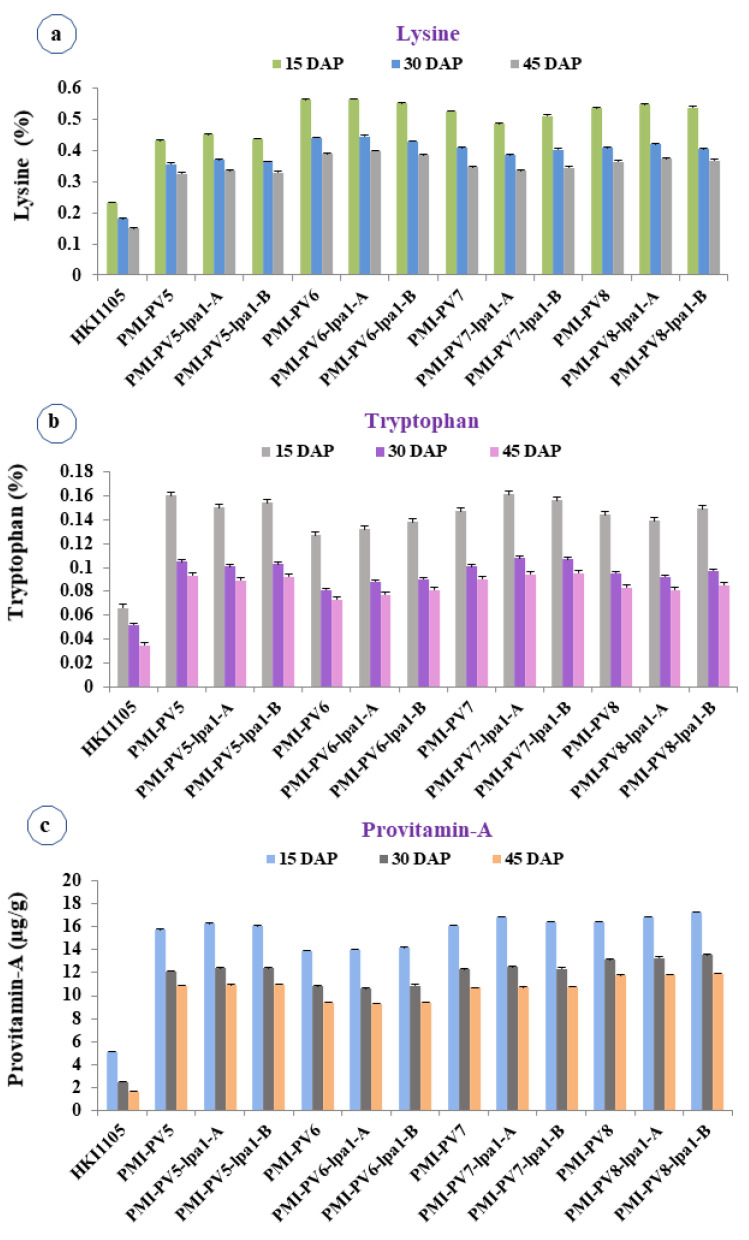
Mean concentration of nutritional parameters at different kernel developmental stages. (**a**) Lysine, (**b**) Tryptophan, (**c**) Provitamin-A. DAP: Days after pollination. Error bars represents the standard error at 5% level of significance.

**Figure 3 plants-12-01745-f003:**
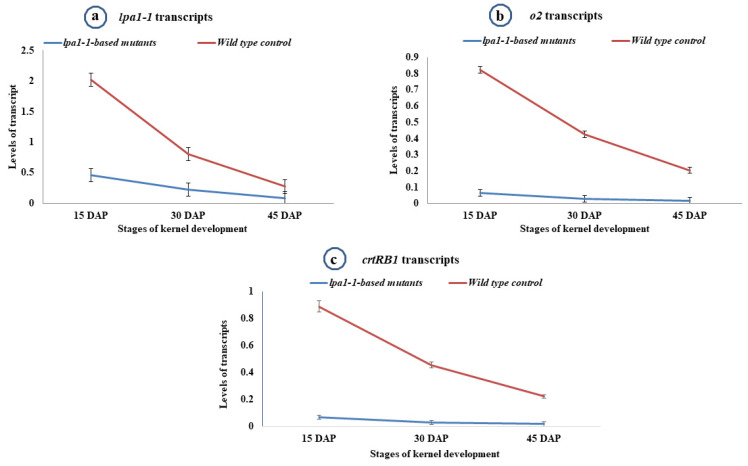
Comparative mean expression of (**a**) *lpa1-1*, (**b**) *o2*, and (**c**) *crtRB1* genes in *lpa1-1*-based mutants (improved) and wild-type (control) inbreds at three stages of kernel development. DAP: days after pollination. Error bars represent the standard deviation.

**Table 1 plants-12-01745-t001:** Combined ANOVA for nutritional quality traits among the genotypes under study.

Source of Variation	*df*	PA	iP	TP	PA/TP	Lysine	Tryptophan	ProA
Replication	2	0.00090	0.00023	0.00158	0.00001	0.00006	0.00001	0.00100
Genotype (G)	12	2.47800 **	0.57529 **	0.69676 **	0.13798 **	0.04815 **	0.00305 **	75.62000 **
Days after pollination (DAP)	2	3.55780 **	0.53805 **	1.39162 **	0.19262 **	0.22726 **	0.03695 **	259.14900 **
G × DAP	24	0.00470 **	0.04631**	0.06084 **	0.00099 **	0.00092 **	0.00010 **	0.33000 **
Error	76	0.00170	0.00118	0.00352	0.00009	0.00003	0.00002	0.00800

** Significant at *p* < 0.01.

**Table 2 plants-12-01745-t002:** Combined ANOVA for transcript levels of *lpa1-1, crtRB1* and *o2* genes.

Source of Variation	*df*	*lpa1-1*	*crtRB1*	*o2*
Replication	2	0.16210	0.00062	0.00002
Genotype (G)	12	1.45110 **	0.05380 **	0.13870 **
Days after pollination (DAP)	2	8.40920 **	0.00901**	0.09144 **
Genotype × DAP	24	0.39270 **	0.00151 **	0.01889 **
Error	76	0.05520	0.00061	0.00044

** Significant at *p* < 0.01.

**Table 3 plants-12-01745-t003:** Transcript levels of *lpa1, crtRB1* and *o2* genes among mutant and wild-type genotypes.

Genotypes	*lpa1-1*			*crtRB1*			*o2*		
	15 DAP	30 DAP	45 DAP	15 DAP	30 DAP	45 DAP	15 DAP	30 DAP	45 DAP
PMI-PV5	2.091	0.880	0.276	0.032	0.025	0.016	0.061	0.027	0.012
PMI-PV5-*lpa1*-A	0.464	0.214	0.099	0.030	0.026	0.017	0.062	0.027	0.016
PMI-PV5-*lpa1*-B	0.473	0.222	0.075	0.037	0.028	0.014	0.065	0.024	0.012
PMI-PV6	1.816	0.715	0.252	0.035	0.025	0.013	0.083	0.029	0.019
PMI-PV6-*lpa1*-A	0.421	0.225	0.091	0.030	0.027	0.016	0.079	0.029	0.017
PMI-PV6-*lpa1*-B	0.469	0.241	0.087	0.032	0.026	0.010	0.077	0.031	0.018
PMI-PV7	2.045	0.836	0.264	0.033	0.026	0.013	0.056	0.028	0.011
PMI-PV7-*lpa1*-A	0.429	0.205	0.064	0.031	0.028	0.011	0.062	0.026	0.019
PMI-PV7-*lpa1*-B	0.451	0.219	0.074	0.030	0.029	0.018	0.057	0.027	0.016
PMI-PV8	2.121	0.792	0.295	0.029	0.028	0.018	0.067	0.025	0.020
PMI-PV8-*lpa1*-A	0.494	0.238	0.085	0.033	0.028	0.015	0.069	0.030	0.019
PMI-PV8-*lpa1*-B	0.475	0.202	0.073	0.032	0.029	0.014	0.066	0.027	0.017
HKI1105	2.105	0.870	0.288	0.387	0.314	0.209	0.821	0.427	0.205
CD at 5%	0.599	0.289	0.086	0.578	0.044	0.007	0.041	0.014	0.019
SE	0.205	0.099	0.029	0.020	0.015	0.002	0.014	0.040	0.006

DAP: Days after pollination.

**Table 4 plants-12-01745-t004:** Correlation between gene expression and nutrient accumulation at different stages of kernel development.

S. No.	Traits	DAP	Correlation Coefficient (r)
** *lpa1-1* **			
1	PA	15 DAP	0.99 ***
2	PA	30 DAP	0.98 ***
3	PA	45 DAP	0.98 ***
4	iP	15 DAP	−0.90 ***
5	iP	30 DAP	−0.97 ***
6	iP	45 DAP	−0.98 ***
7	PA/TP	15 DAP	0.98 ***
8	PA/TP	30 DAP	0.98 ***
9	PA/TP	45 DAP	0.98 ***
** *o2* **			
1	Lysine	15 DAP	−0.84 ***
2	Lysine	30 DAP	−0.91 ***
3	Lysine	45 DAP	−0.91 ***
4	Tryptophan	15 DAP	−0.92 ***
5	Tryptophan	30 DAP	−0.85 ***
6	Tryptophan	45 DAP	−0.91 ***
** *crtRB1* **			
7	ProA	15 DAP	−0.94 ***
8	ProA	30 DAP	−0.94 ***
9	ProA	45 DAP	−0.94 ***

*** Significance level (*p* < 0.001); DAP: days after pollination.

## Data Availability

The relevant data and additional information are available in the Appendix A.

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
