# Peer review of "Expression Dynamics of lpa1 Gene and Accumulation Pattern of Phytate in Maize Genotypes Possessing opaque2 and crtRB1 Genes at Different Stages of Kernel Development"

_plants, 2023, doi:10.3390/plants12091745_

Round 1
Reviewer 1 Report
The manuscript entitled "Expression dynamics of lpa1 gene and accumulation pattern of phytate in biofortified maize possessing opaque2 and crtRB1 genes at different stages of kernel developments" is the first comprehensive study on lpa1-1 expression in biofortified maize inbreds during different kernel development stages. The information generated here offers great potential for understanding the dynamics of phytic acid regulation in maize. The study is weel designed, clearly written, the tables and figures are clear and the conclusions are adjusted to the results obtained. So, I recommend its publications in the present form
Author Response
We sincerely thank the Reviewer for the constructive suggestions to improve the manuscript.
Please see the attachment.

Reviewer 2 Report
The authors quantify the expression level during grain filling of three mutant genes in a number of maize lines which combines three mutations that should provide nutritional improved seeds by 1) reducing phytic acid, 2) increase lysine and 3) increase in pro-vitamin A. The plants were grown in the field 2022 with two replicates and randomized, although only one growth season was carried out it is very much appreciated that field grown material is used. The lines address an important challenge in cereal crops, to improve bioavailability of Fe and Zn by reducing the anti-nutritional factor phytic acid, improve grain protein with lysin and increase the amount of pro-vitamin A and it seems to be an excellent collection of pre-breeding maize lines. The study is missing determination of the central micro-nutrients Fe and Zn and hence it can not be evaluated if the triple mutant lined have been improved. To efficiently minimise Fe and Zn absorption inhibition, the molar ratios for phytate : Fe should be less than 1.0 (Hurrell, 2004), phytate : Zn molar ratio less than 15 (World Health Organization, 2004) see eventually introduction in Maternal and Child Nutrition (2013), 9 (Suppl. 1), pp. 47–71. The phytic acid : Total-P ratio is not very helpfull for evaluating the bioavailability of Fe and Zn. Also grain protein should be determined, a standard would be total-N and conversion into protein.Also the number of seeds harvested per plant and average-seed weight would be important to present and discuss in context of supplied high quality maize for consumption. In the end of the manuscript the authors briefly introduce the concept of developing green maize cob into a vegetable for consumption, rather than the harvesting the mature grains. This idea could be introduced at the beginning to better understand why mature grains were on not included in the study. The study is otherwise well conducted using state-of-the-art methods.
The authors may consider using seed or grain or kernel; define the term used "biofortified maize"; "estimation" of quality parameters, should be "determination"; it is difficult to understand how immature grain at 15 DAP, 30 DAP and 45 DAP can be dryed down to below 15% water at a room temperature of only 25 C.
Author Response
We sincerely thank the Reviewer for the constructive suggestions to improve the manuscript. All the suggestions have been incorporated in the manuscript and the revised manuscript is submitted for your kind consideration.
Please see the attachment.

Reviewer 3 Report
Dear Authors,
I reviewed your article titled in (Expression dynamics of lpa1 gene and accumulation pattern of phytate in biofortified maize possessing opaque2 and crtRB1 genes at different stages of kernel development).
In general, this research provides interesting and valuable information that can be widely used, especially in the field of breeding and the development of new varieties of maize.
The article is written very well and organized. There are some minor comments that could be useful to improve the quality of your article, which I have mentioned in attached pdf file because there are no lines number in the file.
Abstract and Introduction: The abstract and introduction are written well and have enough literatures that related to the subject. Some minor comments in attached file.
Materials and methods:
All methods are enough described. The design of experiment is correct.
Result and Discussion
This section overall is very well written. The language used is clear, concise, and sufficiently descriptive with a logical flow between statements.
Conclusion
This section overall is well.

Author Response

(The authors gave the same response as above.)

Round 2
Reviewer 2 Report
At the very beginning of the abstract and throughout the manuscript the focus is on "Micronutrient malnutrition, which is majorly caused due to consumption of food deficient in iron (Fe) and zinc (Zn) is a global health concern. Fe and Zn in maize kernels are less bioavailable in human gut due to higher phytic acid (PA) that chelates the minerals." see line 15-17. But the concentration of Fe and Zn is NOT determined in this study of the triple mutant maize lines and therefor it is NOT appropriate to make such claim without experimental support. We do not know if the mutant lines also have lower concentration of Fe and Zn and in which case a reduction in phytic acid will not have an impact because there will be too little Fe and Zn in the kernel anyway.
The second goal was to reduce phytic acid by help of mutations, and this was achieved and experimentally verified and the reduction of phytic acid was quantified.
The third goal was to improve protein quality by help of mutations that would increase the essential amino acids lysine and tryptophan in the kernel. Here some attempt to quantify lysine and tryptophan is made, however, the authors "estimate" which means it is an approximate calculation of the quantity of these to two important amino acids, which seems to be a weakness of the study.
It is a concern that the authors uses estimation methods rather than determination of key amino acids in the mutant kernels and they do not measure total protein in the kernel.
They propose to breed for immature kernels at 45 days after pollination, but there is no detection of minerals that shows Fe and Zn is present at this stage in any significant amount relevant for nutrition.
It is appreciated that field grown material is used which makes it more realistic for assessing agronomic performance, for maize thousand kernel weight (TKW) is very strong reference to compare the performance, unfortunately the authors do not include this. Also protein-% is very relevant in particular because the author has this as the third goal. When developing mutant lines a yield penalty is expected, but without measuring relevant agronomic parameters it is impossible discuss and valuate.
Because of the missing determination / quantification of the micronutrients Fe and Zn, which both are central for biofortified maize the study is preliminary.
Author Response
We sincerely thank you for the suggestions for the improvement of the manuscript. The suggestions have been addressed with thorough revision of the manuscript incorporating the suggestions. Point wise response is given below:
Reviewer Report round 2:
Comment 1: At the very beginning of the abstract and throughout the manuscript the focus is on "Micronutrient malnutrition, which is majorly caused due to consumption of food deficient in iron (Fe) and zinc (Zn) is a global health concern. Fe and Zn in maize kernels are less bioavailable in human gut due to higher phytic acid (PA) that chelates the minerals." see line 15-17. But the concentration of Fe and Zn is NOT determined in this study of the triple mutant maize lines and therefor it is NOT appropriate to make such claim without experimental support. We do not know if the mutant lines also have lower concentration of Fe and Zn and in which case a reduction in phytic acid will not have an impact because there will be too little Fe and Zn in the kernel anyway.
Response 1: The lines 15-17 have been removed and only the relevant text has been included in the manuscript, as per the suggestion.
Comment 2: The second goal was to reduce phytic acid by help of mutations, and this was achieved and experimentally verified and the reduction of phytic acid was quantified.
Response 2: Thank you very much for your kind appreciation of the work.
Comment 3: The third goal was to improve protein quality by help of mutations that would increase the essential amino acids lysine and tryptophan in the kernel. Here some attempt to quantify lysine and tryptophan is made, however, the authors "estimate" which means it is an approximate calculation of the quantity of these to two important amino acids, which seems to be a weakness of the study.
Response 3: The concentration of the amino acids viz. lysine and tryptophan were determined through Ultra-high performance liquid chromatography (UHPLC) in the present study, which is the most accurate way of determining the concentration of amino acids from grain samples. For more clarity to the readers, we have changed the term “estimate” to “determination” in the manuscript, as per the suggestion.
Comment 4: It is a concern that the authors uses estimation methods rather than determination of key amino acids in the mutant kernels and they do not measure total protein in the kernel.
Response 4: In the present study, we have determined the concentration of amino acids and carotenoids using HPLC, the most robust way to determine these compounds from the grain samples. The term “estimate” has been changed to “determination” in the manuscript as per the suggestion. The opaque2 gene only affects the concentration of amino acids viz. lysine and tryptophan but not the quantity of the protein; and in the present study we have determined precisely the concentration of lysine and tryptophan in the mutant and the wild type genotypes.
Comment 5: They propose to breed for immature kernels at 45 days after pollination, but there is no detection of minerals that shows Fe and Zn is present at this stage in any significant amount relevant for nutrition.
Response 5: As suggested in Comment 1, the mention of Fe and Zn has been removed from the entire manuscript.
Comment 6: It is appreciated that field grown material is used which makes it more realistic for assessing agronomic performance, for maize thousand kernel weight (TKW) is very strong reference to compare the performance, unfortunately the authors do not include this. Also protein-% is very relevant in particular because the author has this as the third goal. When developing mutant lines a yield penalty is expected, but without measuring relevant agronomic parameters it is impossible discuss and valuate.
Response 6: The main focus of the study was to assess the expression pattern of the lpa1-1, o2 and crtRB1 genes and their effect on nutrient accumulation (phytic acid, provitamin-A, lysine and tryptophan) in the triple mutant background as compared to their wild type genotypes. The lpa1-1 introgressed lines developed in our lab and characterized by Ragi et al. (2021), didn’t notice any yield penalty. Similar was the report by Raboy et al. (2001) where no yield penalty was observed using the lpa1-1 mutants. Here, we could establish the expression pattern of the genes and their effect on nutrient accumulation. The opaque2 gene affects the concentration of amino acids viz. lysine and tryptophan but not the quantity of the protein; and in the present study we have determined precisely the concentration of lysine and tryptophan in the mutant and the wild type genotypes.
Comment 7: Because of the missing determination / quantification of the micronutrients Fe and Zn, which both are central for biofortified maize the study is preliminary.
Response 7: The primary objective of the study was to understand the expression dynamics of lpa1-1, crtRB1 and o2 genes on the accumulation of phytic acid, carotenoids and amino acids (lysine and tryptophan), respectively. The study is very unique and novel, as we have developed and analysed the triple mutant (lpa1lpa1/crtRB1crtRB1/o2o2) combinations as compared with wildtype, which is the first of kind report globally.
We have determined the concentration of phytic acid using spectrophotometer and the carotenoids and amino acids using HPLC, the most robust method widely followed for determining the concentration of these compounds. The study clearly established the effect of these genes on the concentration of these nutritional compounds and the relationship among them at different stages of kernel development. Thus, the information generated here offers great potential for comprehending the dynamics of phytic acid regulation in maize. Since, we did not determine the concentration of Fe and Zn in the manuscript, we have removed the mention of Fe and Zn from the entire manuscript.